# Do High Doses of Multiple Antibiotics Loaded into Bone Cement Spacers Improve the Success Rate in Staphylococcal Periprosthetic Joint Infection When Rifampicin Cannot Be Employed?

**DOI:** 10.3390/antibiotics13060538

**Published:** 2024-06-10

**Authors:** Lourdes Prats-Peinado, Tanya Fernández-Fernández, Miguel Márquez-Gómez, José Antonio Matas-Diaz, Mar Sánchez-Somolinos, Sofía de la Villa-Martínez, Javier Vaquero-Martín, Pablo Sanz-Ruiz

**Affiliations:** 1Department of Orthopedic Surgery, General University Hospital Gregorio Marañón, 28007 Madrid, Spain; lourdes.prats@salud.madrid.org (L.P.-P.); tanya.fernandez@salud.madrid.org (T.F.-F.); miguel.marquez@salud.madrid.org (M.M.-G.); joseantonio.matas@salud.madrid.org (J.A.M.-D.); jvaquero@salud.madrid.org (J.V.-M.); 2Department of Microbiology, General University Hospital Gregorio Marañón, 28007 Madrid, Spain; mssomolinos@salud.madrid.org (M.S.-S.); sofiadela.villa@salud.madrid.org (S.d.l.V.-M.); 3Surgery Department, School of Medicine, Complutense University of Madrid, 28040 Madrid, Spain

**Keywords:** ALBC, periprosthetic joint infection, mobile spacer, high-dose antibiotic bone cement, staphylococcal infection

## Abstract

Rifampicin is one of the mainstays in treating staphylococcal prosthetic joint infection (PJI). However, discontinuation due to intolerance, drug interactions, and adverse events is common. Two-stage revision surgery remains the gold standard, with the number of revision arthroplasties steadily increasing. This study aims to evaluate the effectiveness and safety of a novel two-stage revision protocol for staphylococcal prosthetic joint infection (PJI) utilizing bone cement spacers loaded with multiple high doses of antibiotics. Additionally, it seeks to analyze outcomes in patients ineligible for rifampicin treatment. A retrospective review of 43 cases of staphylococcal hip and knee prosthetic joint infections (PJIs) from 2012 to 2020 was conducted. In all instances, a commercial cement containing 1 g of gentamicin and 1 g of clindamycin, augmented with 4 g of vancomycin and 2 g of ceftazidime, was employed to cast a spacer manually after thorough surgical debridement. We report an eradication rate of 82%, with no significant differences observed (*p* = 0.673) between patients treated with (84%, *n* = 19) and without rifampicin (79%, *n* = 24). There were no disparities in positive culture rates (7%), spacer replacement (18%), or survival analysis (*p* = 0.514) after an average follow-up of 68 months (range 10–147) in the absence of systemic toxicity and surgical complications superimposable to those previously reported. In conclusion, two-stage revision with local high doses of ceftazidime, vancomycin, gentamicin, and clindamycin demonstrates high effectiveness in treating staphylococcal PJIs. Notably, systemic rifampicin does not influence the outcomes. This protocol, with multiple high doses of antibiotics loaded into the bone cement spacer, is presented as a viable and safe alternative for patients unsuitable for rifampicin treatment.

## 1. Introduction

Over the past years, the demand for primary hip and knee prosthetic joint replacement has risen due to improved life expectancy and functional requirements in growing elderly populations. One of the most devastating complications is prosthetic joint infection (PJI), occurring in 1–2% of primary cases and in up to 20% of revision surgeries [1]. However, it is expected that we will face an increase in PJI and revision arthroplasties in the foreseeable future [2]. In this scenario, it is of utmost importance to develop appropriate strategies that encompass surgical and antimicrobial treatment within a multidisciplinary approach while simultaneously addressing the emergence of bacterial resistance.

The most frequent causative pathogens of PJI are *Staphylococci aureus* and *Staphylococci epidermidis*, but the percentage of methicillin-resistant strains varies considerably between continents (12% in Europe vs. 48% in the USA) [3]. The development of a mature polymer-based matrix endows these bacteria with reduced susceptibility to the immune system and antimicrobials, rendering staphylococcal PJIs challenging to treat and eradicate [4]. Since the introduction of rifampicin, there has been a substantial increase in the success rate of treating such infections due to its antibiofilm effect. The available evidence from in vitro animal studies and patients undergoing DAIR (debridement, antibiotics, and implant retention) following orthopedic device-related infection supports the excellent efficacy of this antimicrobial against biofilm-forming staphylococci [5,6]. However, data regarding rifampicin efficacy in chronic staphylococcal PJI are limited to observational studies, and its overall effectiveness for patients undergoing two-stage revision arthroplasty remains uncertain to this day [7,8,9]. In addition to biofilm, resistance mechanisms have been described in recent years that partly explain the difficulty of eradicating staphylococcal PJI. Specifically, small colony variants (SCVs) serve as a form of intracellular latent resistance [10]. The ideal anti-SCV therapy would require an antimicrobial that can penetrate a host cell and maintain its activity. When dealing with intracellular *Staphylococcus aureus* SCVs, clinicians are limited to a handful of antibiotic choices, such as rifampicin and specific glycopeptides [11]. This suggests the suitability and efficacy of rifampicin for the treatment of chronic infections.

However, rifampicin is often discontinued due to adverse events or toxicity, as noted in previous studies related to tuberculosis and implant-associated infections. Gastrointestinal intolerance is the most commonly reported side effect, but drug–drug interactions may also occur since rifampicin is a potent inducer of cytochrome P450 [12]. Mainly, elderly patients are prone to discontinue rifampicin in up to 50% of cases [13]. Given the more prolonged survival of primary hip and knee prostheses and the increased risk of septic complications related to aging, treatment alternatives for patients unsuitable for rifampicin treatment need to be explored. However, research on tailored strategies remains limited [9,14].

The use of local antibiotics is one of the recommended approaches to treat PJI, with antibiotic-loaded bone cement being the most widely used [15,16,17]. In particular, dual-antibiotic-loaded bone cement (vancomycin + aminoglycoside) has been found to be more effective and cost-efficient than single antibiotics for the treatment of PJI [18]. However, in the best scenario, only 10% of the vancomycin added to PMMA will be eluted. To enhance elution, various alternatives have been proposed, among which the addition of a second, more hydrophilic antibiotic has proven to be one of the most effective strategies. Paz et al. showed that the addition of cefazoline to vancomycin-loaded bone cement increases vancomycin elution more than augmenting the quantity of vancomycin itself [15]. We hypothesized that patients who are ineligible for systemic rifampicin may benefit from enhancing local treatment through the elution of high doses of antibiotics from the spacer. These high concentrations can only be achieved by manually adding the antibiotics to PMMA and casting a hand-made spacer.

This study aims to (1) analyze the efficacy and safety of a two-stage revision protocol for staphylococcal periprosthetic infection utilizing a multiple high-dose antibiotic-loaded cement (MHDALC) spacer with ceftazidime, vancomycin, clindamycin, and gentamicin; (2) analyze and compare the outcomes in patients unsuitable for systemic rifampicin treatment; and (3) analyze the systemic and mechanical safety of the manually casted MHDALC spacer.

## 2. Results

### 2.1. Population Included

A total of 43 patients (18 males [42%], mean age of 70 years [IQR 66–76]) with confirmed staphylococcal PJI affecting total knee arthroplasty (TKA) (*n* = 24, 55%), total hip arthroplasty (THA) (*n* = 17, 39.5%) and hip hemiarthroplasty following femoral neck fracture (HHA) (*n* = 2, 4.5%) were included. They were categorized according to systemic antibiotic treatment, resulting in 55% of the cases receiving a rifampicin-free regimen. In terms of baseline comorbidities, diabetes (*n* = 7, 16%; *p* = 0.911), oral anticoagulant therapy (*n* = 7, 16%; *p* = 0.631), and previous heart disease (*n* = 9, 23%, *p* = 0.679) were the most prevalent. The mean prosthesis survival time before the diagnosis of infection was 142 weeks (IQR 8–144 weeks), although in rifampicin-treated patients, the trend was toward a shorter interval (mean 85 weeks, IQR 8–89). Notably, 37.2% (*n* = 16, *p* = 0.521) of the PJIs followed revision arthroplasties (aseptic revision of primary components), and 35% (*n* = 15) were referred from another center for definitive treatment. These patients were prone to be treated without rifampicin. Previous DAIR had failed in 16.3% of cases (RFP regimen 10.5% [*n* = 2], non-RFP regimen 20.8% [*n* = 5], *p* = 0.294). The patients’ baseline characteristics are summarized in Table 1.

Most variables were homogeneously distributed between the two groups. Significant differences between baseline cohorts regarding chronic kidney disease (*p* = 0.03) and polymicrobial infection (*p* = 0.013) were found, both affecting the cohort treated without rifampicin to a greater extent.

Regarding microbiological findings, *Staphylococcus epidermidis* was the main causative pathogen identified in 44% of the patients (*n* = 18). *Staphylococcus aureus* was isolated in 27.5% (*n* = 12), other coagulase-negative staphylococci (CNS: *S. lugdunensis* and *S. capitis*) in 7% (*n* = 3), and staphylococcus involved in polymicrobial infection caused 21% (*n* = 10) of the PJIs. Methicillin-resistant strains amounted to 46% (*n* = 20), and resistance to vancomycin or rifampicin was noted in 4.6% and 2.3%, respectively. A detailed display of microbiological findings and resistances can be found in Table 2.

### 2.2. Patient Management

Patients received antibiotic treatment for a mean of 11.5 weeks (SD 5.9) following the first stage (RFP regimen 11.8 weeks vs. non-RFP regimen 11.4 weeks; *p* = 0.837), and the mean time to reimplantation was 14.6 weeks (RFP regimen 13 weeks vs. non-RFP regimen 15 weeks; *p* = 0.079). Following our treatment algorithm, levofloxacin and cotrimoxazole, both in combination, were the preferred regimens (45.8% of patients, *n* = 11), followed by daptomycin-based regimens (33%, *n* = 8). For methicillin-resistant strains, cotrimoxazole and levofloxacin were the most prescribed agents, each accounting for 33% of prescriptions. Appendix A contains a more detailed description of the treatment strategies and outcomes of patients unsuitable for rifampicin treatment. Spacer design was comparable in both groups, with 72% being articulated (RFP 79% vs. Non-RFP 67%; *p* = 0.373).

### 2.3. Outcomes

Based on the definition of therapeutic failure established for this study, there was no significant difference (*p* = 0.673) in the eradication rate between the cohort treated with (84%) and without rifampicin (79%), with an overall satisfactory outcome of 82% after a mean follow-up time of 68 months (range 10–147 months). The main cause of treatment failure was recurrence (11%, *n* = 5), and superinfection was related to 7% of the unfavorable outcomes. The microorganisms isolated in cases of superinfection were MSSA, *Candida parapsilosis*, and multidrug-resistant *Pseudomonas aeruginosa*. In contrast, infection recurrence occurred in 4 polymicrobial and 1 MRSE infection. Only 12.5% of these patients underwent suppressive therapy. The management and treatment outcomes are summarized in Table 3. After the exclusion of patients treated with two-stage revision following tumoral reconstructive PJIs, infection eradication was achieved in 86% of cases (RFP regimen 87.5% [*n* = 14] vs. non-RFP regimen 85.7% [*n* = 18], *p* = 0.873). A Kaplan–Meier plot shows the time to failure for both groups in Figure 1, with no statistical differences found in survival (log rank *p* = 0.514). In multivariate analysis, polymicrobial staphylococcal infection (OR 14.3, CI 1.05–203) and septic spacer replacement (OR 14.25, CI 1.42–142.1) were the only two variables related to therapeutic failure. Time from index prosthesis implantation showed a strong association with failure (*p* = 0.072). Table 4 summarizes the results of both univariate and multivariate analyses assessing therapeutic failure.

Spacer cultures after sonication were positive in three patients (RPF regimen 5% [*n* = 1] vs. non-RFP regimen 8.3% [*n* = 2], *p* = 0.695), and notably, treatment failure occurred in all cases. Isolated microorganisms were indicative of MRSE persistent infection in 3/3, but superinfection by *Candida parapsilosis* and multiresistant *Pseudomonas aeruginosa* was also identified in 2/3 samples, respectively.

In terms of safety, major surgical complications occurred in 14% (*n* = 6) of patients, including spacer breakage (*n* = 1, 2.3%), spacer dislocation (*n* = 4, 9.3%), and intraoperative peri-implant fracture (*n* = 1, 2%). This rate was higher among hip spacers (26% vs. 5.8% of knee spacers). Spacer replacement was performed due to persistent infection in 11.6% of cases, and a decrease in the eradication rate to 40% was observed in this subgroup. Four patients underwent spacer replacement because of mechanical complications (luxation or spacer fracture), contributing to a cumulative 21% (*n* = 9) of spacer revision surgeries. The incidence of therapeutic failure, 1 out of 4 cases (25%), was comparable to that of the rest of the cohort. The main primary and secondary outcomes are displayed in Figure 2.

An overview of surgical complications can be found in Table 5. The cohort that did not receive rifampicin exhibited higher rates of medical complications (*n* = 12, 50%). Despite its potential clinical relevance, this outcome did not reach statistical significance in comparison to the RFP group (*n* = 6, 32%; *p* = 0.224). The prevailing complications included anemia requiring transfusion in 9 patients (20%), acute prerenal kidney failure in 6 patients (14%), urinary tract infection in 2 patients (4.6%), and decompensation of congestive heart failure in 2 patients (4.6%). Additionally, there was one recorded decompensation of liver failure and encephalopathy (not treated with RFP), one occurrence of late nickel allergy that led to aseptic revision arthroplasty, and one case of non-drug-associated reactive thrombocytopenia. No cases of spacer removal secondary to systemic complications (renal failure or others) were found. The mean hospitalization time amounted to 58 days (SE 9.5) and was substantially longer (*p* = 0.219) in the group treated without rifampicin (76 days mean [SE 16]) versus 37 days (SE 6.4) in the non-RFP cohort.

## 3. Materials and Methods

### 3.1. Study Design and Participants

This is a retrospective, single-center, observational study conducted in an academic tertiary hospital (Gregorio Marañón General University Hospital). Enrolled patients were diagnosed with staphylococcal infection following total knee or hip replacement and underwent two-stage revision arthroplasties between 2012 and 2020. Participants were selected from a prospectively collected database, and clinical data were obtained from a retrospective review of electronic clinical records. The diagnosis of PJI was based on the 2013 International Consensus Meeting definition [20]. The detailed protocol was applied in all cases, and surgeries were performed by the senior surgeon. We excluded patients who did not undergo our specific protocol, did not complete the second stage, were lost to follow-up during the first year, or discontinued rifampicin treatment after its implementation, as presented in Figure 3. A total of 43 patients with staphylococcal hip and knee PJI fulfilled the inclusion and exclusion criteria. Among these, 19 patients were treated with a rifampicin-based regimen and 24 patients with selected systemic antibiotics that did not include rifampicin.

### 3.2. Ethical Statement

The study was approved by the Scientific Ethics and Drug Research Committee of Gregorio Marañón University General Hospital. In accordance with Spanish legislation regarding retrospective observational studies, all patients were provided with written information about the study and given the option to decline participation. However, the requirement for written informed consent was waived.

### 3.3. Treatment Protocol

Patients underwent previously standardized two-stage revision surgery with high doses of local antibiotics and received systemic antibiotic treatment for staphylococcal PJI. All prosthetic material and intra-articular foreign bodies were removed during the first stage of surgery. Intensive surgical and mechanical debridement of devitalized tissues, synovectomy, and reaming of the endomedullary cavity were subsequently performed. This was followed by low-pressure pulsed irrigation with 3 L of physiological saline (PS) 0.9% to reduce microbial load and chemical debridement with 3% step-by-step hydrogen peroxide and 50% diluted povidone-iodine. Samples for microbiological analysis were collected based on our institutional protocol and sent for aerobic or anaerobic, fungal, and acid-fast bacilli culture using solid media and broth. In brief, before skin incision (intra-articular needle puncture), one fluid sample was obtained; after that, during initial debridement, five tissue samples were collected from three standardized surgical sites: synovial membrane, periarticular bone biopsy, and the membrane between the implant and the bone. The bone biopsy was obtained from the cotyloid cavity and femoral endomedullary cavity in cases of hip revision surgery and from the femoral surface and endomedullary cavity of the tibia in cases of knee revision surgery. In all cases, prosthetic material was sent for sonication. Additional cultures were taken on a case-by-case basis in areas that seemed to have a high yield. Further details of the culture preparation and technique can be found in Appendix A.

Once the surgical site was prepared for the spacer placement, it was rinsed with low-pressure pulsed lavage (3 L of PS 0.9%). The surgical equipment was wholly replaced, including redraping over the contaminated drapes and exchanging the instrument set used for hardware removal. In all cases, the commercial cement Copal G + C^®^ (Heraeus Medical LLC, Yardley, PA, USA), containing 1 g of gentamicin and 1 g of clindamycin per 40 g of cement, was employed, to which 4 g of vancomycin and 2 g of ceftazidime were manually added while mixing the powdered interphase at atmospheric pressure. Two bags of this commercial cement were used for each case, amounting to 80 g of cement, 8 g of vancomycin, 4 g of ceftazidime, 2 g of clindamycin, and 2 g of gentamicin. The spacer was cast manually during the cement-working phase and implanted to complete the first revision stage. The decision to utilize either an articulated or static spacer was at the surgeon’s discretion, considering factors such as the resulting bone stock, previous limb function, extensor mechanism evaluation, and the patient’s ability to cooperate postoperatively.

The knee-articulated cement spacer was molded by hand if minor bone defects were present (Figure 4A). For patients with significant bone defects or lacking collateral ligament support, we utilized a relatively constrained ball-and-socket spacer, following the technique described by MacAvoy and Ries. In this technique, a convex femoral spacer is shaped using the bulb portion of a rubber irrigation syringe as a mold. The tibial spacer is formed by placing cement in the doughy phase on the proximal tibia and using the bulb washing syringe to create a hemispherical concavity. Any additional cement is used to fill bony defects, which are addressed when placing the spacer.

In hip revisions (Figure 4B), the femoral head was also fashioned using the mold of a washing syringe, inverted to utilize the smooth part on the cephalic screw or Kuntscher´s nail, as we have described in previous publications [21]. After confirming adequate acetabular congruency, the nail was coated with cement, with attention paid to avoiding excessive material in the trochanteric zone to prevent complications during closure. Small perforations were made in the cement layer at the trochanteric level for subsequent reattachment of the abductor and psoas muscles. Finally, the nail was entirely coated with cement and placed intramedullarily.

Except for those patients with signs of sepsis, antibiotic treatment was started empirically after the removal of the prosthetic material. The administration of parenteral meropenem 1 g/8 h and teicoplanin 600 mg/24 h (first day every 12 h) was the standard regimen for the first week. In the case of beta-lactam allergy, ciprofloxacin 400 mg/12 h was the alternative. This empiric treatment was later tailored according to the results of the cultures in consultation with a musculoskeletal infectious disease specialist. The first-line combination was oral levofloxacin 500 mg/day + oral rifampicin 600 mg/day for six weeks. In cases of resistance or intolerance, alternatives to levofloxacin were oral clindamycin 300 mg/8 h or cloxacillin 1 g/6 h. For infections caused by methicillin-resistant *Staphylococcus aureus* (MRSA) or methicillin-resistant coagulase-negative *Staphylococcus* (MRCNS), the choice was cotrimoxazole 800 mg/12 h or linezolid 600 mg/12 h associated with rifampicin 600 mg/day orally. One week of parenteral antibiotic treatment was accomplished before switching to oral therapy.

For those patients with previous hepatic insufficiency, at high risk of interaction with other drugs (e.g., oral anticoagulants), rifampicin resistance, or at high risk of treatment discontinuation, the antimicrobial therapeutic regimen was modified according to microbiological criteria and rifampicin was not added to treatment.

Clinical and analytical follow-up was performed every 2–4 weeks at microbiology and orthopedic consultations. The decision to undergo reimplantation was based on assessing serum inflammatory marker trends (CRP and ESR) and evaluating wound healing. Patients were managed with an antibiotic-free period ranging from 2 to 3 weeks before reimplantation to verify that the infection was successfully eradicated. The standard procedure of aspiration before reimplantation was omitted. In patients showing signs of potential persistent infection, such as impaired wound healing, intraoperative purulence, or mechanical complications, we performed a spacer exchange to repeat debridement and introduce a new load of antibiotics. Likewise, surgical, chemical, and mechanical debridement were repeated in spacer exchanges prompted by mechanical failure.

The second stage was conducted under targeted prophylactic antibiotics administered at least 30 min before skin incision. An aggressive debridement was performed after spacer removal, and at least five deep tissue samples were collected for culture, including the spacer sent for sonication (following the same protocol as in the first stage). The significance attributed to positive cultures was thoroughly discussed by our multidisciplinary unit (orthopedic surgeons and microbiologists specializing in musculoskeletal infections), considering both the number of positive samples and the isolated microorganisms. In cases where no viable microorganisms were identified within a timeframe of 7–10 days, antibiotic administration was ceased. Conversely, when a positive culture matched the previously isolated microorganism, a minimum 6-week course of targeted systemic antibiotics was completed. The presence of a new microorganism in more than one sample was considered indicative of superinfection, and these patients received an additional 6- to 8-week course of tailored systemic antibiotics. In both scenarios, the addition of rifampicin was performed based on the criteria described above for the first stage. If positive cultures were determined to be indicative of contamination, no further antibiotic treatment was administered.

### 3.4. Outcome Assessment

The primary outcome was the eradication of staphylococcal PJI, defined as the absence of signs and symptoms of infection in patients with retained implants during the entire follow-up period, along with a lack of criteria for treatment failure. Failure criteria included (1) reintervention due to septic reasons after completion of the second stage (infection persistence, reinfection, or superinfection), (2) PJI-related death, and (3) the need for suppressive antibiotic therapy. Secondary outcomes comprised spacer exchange due to persistent infection and the incidence of positive spacer cultures. To assess safety, surgical (mechanical) and systemic complications were taken into consideration.

### 3.5. Statistical Analysis

Descriptive analysis was used to reproduce the baseline characteristics of the patients after stratification into cohorts according to rifampicin treatment. Dichotomous variables are presented as percentages, and continuous variables are summarized with the mean, interquartile range (IQR), and standard deviation (SD). Differences between groups were compared using the chi-squared test for categorical variables or Fisher’s exact test when appropriate. The distribution of continuous variables was tested for normality (Kolmogorov–Smirnov test), and the Student’s *t*-test was used when this criterion was met. The Mann–Whitney U test was the non-parametric alternative. To represent differences in treatment outcomes, survival curves were designed following the Kaplan–Meyer method and assessed by the log-rank statistic. Patients who died during follow-up were censored, and those with treatment failure were considered final events according to the criteria presented above. The association of risk factors with therapeutic failure was evaluated by binary logistic regression. Variables with significance or close to significance in the univariate analysis, as well as those previously reported in the literature, were included in the analysis. All statistical tests were 2-sided, and significance was set at the level of alpha = 0.05. Analyses were performed using SPSS v. 25.0 statistical software (SPSS, Chicago, IL, USA).

## 4. Discussion

The utilization of antibiotic-impregnated polymethylmethacrylate (PMMA) bone-cement spacers is acknowledged as the prevailing standard approach for patients with chronic PJIs and is correlated with a decreased occurrence of treatment failures in septic revision protocols [22]. High doses of two different antibiotics (usually vancomycin + one aminoglycoside) are commonly incorporated to provide a wide antibacterial spectrum and synergistic release kinetics and to counteract the selection of bacterial mutants [23]. However, the best combination of antimicrobials and specific dosage is yet unknown [24,25]. In this study, a handmade antibiotic-loaded bone cement (ALBC) spacer mixing 4 g of vancomycin and 2 g of ceftazidime per 40 g of cement already containing 1 g of clindamycin and 1 g of gentamicin achieved notable therapeutic success rates (82%) in 35 of 43 patients with staphylococcal PJI undergoing two-stage hip and knee revision surgery after a 5-year follow-up. Secondarily, we found that infection was eradicated in 79% of the patients who received a systemic antibiotic regimen without rifampicin along with the specified surgical protocol. No differences (*p* = 0.673) were observed compared to the group treated with a rifampicin-based strategy, with an 84% success rate. We report these outcomes after a mean post-discharge surveillance of 68 months (ranging from 10 to 147 months).

Two-stage revision arthroplasty success rates approach 90%, although there is noteworthy variability across reports ranging from 65% to nearly 100%, and the real success appears to be overestimated, especially in the short-term follow-up [26]. This is partly explained by the lack of agreement about what constitutes a successful treatment, even after the publication of Delphi criteria [27]. Heterogeneity between studies regarding the population enrolled or the follow-up period after which the results are registered hinders their comparability [28]. We believe that in PJI, the follow-up time is highly relevant, as therapeutic failure often occurs even years after the completion of reimplantation. This may be due to the presence of small staphylococcal colonies that alter their metabolic state in the form of persistent cells or small colony variants [11].

Aminoglycosides and glycopeptides, specifically gentamicin, vancomycin, and tobramycin, are the most widespread antimicrobial combinations used on ALBC [22]. Concern arises regarding their linked systemic complications, including renal toxicity, bone marrow depression, and allergic reactions, particularly among the elderly. In turn, some cephalosporins exhibit a water-soluble, thermally stable profile, with the potential advantage of decreased systemic toxicity and comparable efficacy. This is the case with ceftazidime, which could be an alternative local treatment for staphylococcal infections. Previous in vitro studies support the effectiveness of vancomycin and ceftazidime against methicillin-sensitive *Staphylococcus aureus* (MSSA), methicillin-resistant *Staphylococcus aureus* (MRSA), *Staphylococcus epidermidis*, *Pseudomonas aeruginosa*, and *Escherichia coli* [29]. When comparing results between in vivo studies, our overall 82% eradication rate utilizing 4 g of vancomycin and 2 g of ceftazidime for staphylococcal PJI is consistent with other ALBC regimens, such as tobramycin (90.4% eradication rate reported by Westrich et al. [30]), tobramycin plus vancomycin (9% treatment failure in Lewis et al. [31]), and gentamicin plus vancomycin (77.8% success over 108 patients reported by Corro et al. [32]), on which more extensive research has been published. Similar results were obtained (85.4% eradication rate) by doubling the dose of ceftazidime to 4 g plus 4 g of vancomycin in knee PJI [33].

In this study, stringent criteria were applied to define therapeutic failure in comparison with other published series [34]. Any reintervention for septic reasons (persistence of infection, superinfection, or reinfection), including DAIR and infection control with suppressive therapy, were considered failures. PJIs following tumoral reconstructive surgeries were included among the selected cases and exhibited a higher failure rate (50%), so outcomes in this subgroup may not be extensible to other patients. If excluded, an overall 86% cure rate was achieved, irrespective of rifampicin treatment. By analyzing 162 cases, Corona et al. revealed an overestimation of 9% (eradication rate from 80.6% to 71.6%) when accounting for patients who retained the spacer without undergoing reimplantation [35]. Previous research on this matter indicates that 17–30% of patients do not undergo definitive reimplantation [36,37]. In our study, only five patients (10.2%) were excluded for this reason, with a homogeneous distribution between the groups. The observed potential overestimation is consistent with previous data and led to a decline in therapeutic success from 82% to 73% when considering this subgroup. However, the selection of systemic antibiotic treatment did not impact reimplantation.

The efficacy of rifampicin in treating periprosthetic staphylococcal infections is widely acknowledged [38]. Biofilm age and low bacterial load are determinants of rifampicin success [6]. In patients with short duration of symptoms and good soft tissue status, the efficacy of rifampicin in staphylococcal PJI has been largely proven, though recently questioned [13,39]. Its use is extended following DAIR and one-stage replacement, but there is insufficient evidence to routinely recommend rifampicin for patients undergoing two-stage arthroplasty, as studies addressing this question have revealed controversial results [8,9,22]. A recent report showed better results after two-stage surgery if rifampicin was used as a systemic treatment, probably based on its ability to eradicate the rest of the preexisting biofilm and its activity against small colony variants [40]. This concern is greatest in older patients, who often show very low tolerance to rifampicin treatment. Among the individuals who met the inclusion criteria in our study, 55% received a rifampicin-free regimen. The listed reasons were hepatic insufficiency, potential interaction with other drugs, high risk of intolerance, and rifampicin resistance. Based on previous data, withdrawal of rifampicin due to adverse events ranges from 14% to 50% [41]. The tendency to prescribe a strategy not containing rifampicin may be explained by the lack of consensus on the role of rifampicin in chronic infections and two-stage revision arthroplasty, and because patients who discontinued rifampin after initiation were also excluded from the study to truly assess the role of this antimicrobial. Lately, a multicentric study conducted by Herry et al. related rifampicin as the sole antibiotic determinant in the cure of *Staphylococcus lugdunensis* PJI after the observational analysis of 111 patients, but the surgical strategy varied from DAIR to one-stage and two-stage replacement [42]. In the Krizsan et al. cohort of 73 patients treated exclusively with two-stage replacement, rifampicin resistance significantly reduced the cure rate of staphylococcal PJI from 92.5% (rifampicin sensitivity) to 60% with resistance [43]. In part, these results may be explained by the fact that therapeutic options against resistant microorganisms are limited or that the appearance of resistance is due to prolonged previous treatments in cases of difficult-to-eradicate PJIs.

To date, no publications have assessed the effect of rifampicin when a specific high-dose combination of antibiotics is implemented in ALBC. If our surgical debridement and local antibiotic protocol is followed, rifampicin does not influence the outcomes of staphylococcal PJI. In addition, only 3/43 (7%) of the cases presented spacer cultures that were positive before reimplantation, which is relevant in terms of efficacy given that positive cultures may be a risk indicator for therapeutic failure [44,45]. This study does not aim to evaluate the efficacy of rifampicin following two-stage revision, but to assess the outcomes of a protocol for patients ineligible for rifampicin-based regimens. Considering our results, the use of ALBC with 2 g ceftazidime + 4 g vancomycin + 1 g gentamicin + 1 g clindamycin in combination with selected systemic antimicrobials is an effective strategy. These findings favor the limited role of rifampicin in two-stage arthroplasty if adequate local antibiotic concentration is obtained, but further research should address this clinical question.

In our series, the utilization of high doses of multiple antibiotics loaded into bone cement has also proven to be a safe alternative to systemic rifampicin in patients who cannot tolerate it or for whom it is contraindicated. We observed non-significative higher medical complications in the non-rifampicin cohort; however, this finding is related to the significantly higher rate of baseline comorbidities present in this group, particularly preexisting severe kidney disease. This increased comorbidity also contributes to a longer length of stay. Notably, despite the higher number of comorbidities and polymicrobial infections, these patients did not exhibit worse outcomes. In addition to possible systemic toxicity, another concern with the use of MHDALC is its influence on the mechanical properties of the spacer. We observed a 14% mechanical complication with the use of MHDALC. Faschingbauer et al. published their results using a handmade spacer in 128 patients [46]. The overall spacer complication rate was 19.6%. In line with their findings, Jones et al. reported mechanical failures in 26% of cases out of 155 patients [47]. In our study, four of these were dislocations, which occurred regardless of the type of ALBC used for the spacer, and only one (3.2%) was a spacer fracture due to a massive proximal femoral bone after the tumoral spacer.

The limitations of this study include its retrospective design and dependence on information obtained from medical records and our database. Furthermore, as our institution is a tertiary referral center, the potential for selection bias cannot be excluded. Some patients had prior treatment at different clinics and were referred for definitive treatment, which has been demonstrated to worsen outcomes, especially in settings of previous irrigation, debridement, and prosthesis retention. In fact, 37% of PJI cases occurred on previously revised prostheses. Our study did not consider the amount of implanted cement due to the challenging nature of weighing each spacer in a clinical setting. For instance, the MHDALBC spacer was customized to match the joint size, making it impractical to measure its weight accurately. Instead, we relied on the average standardized concentration of hand-mixed antibiotics per cement pack to represent the spacer’s antibiotic content and analyze the impact of additional antibiotics. While antibiotic elution was undoubtedly influenced by the antibiotic amount, Duey et al. highlighted that antibiotic release was correlated with the specimen’s surface area rather than its volume [48]. Finally, although the cohort in this study is relatively large, some predictive variables may not reach significance because of the limited sample size. Although a specific treatment protocol established by our center has been used, there is still controversy regarding the optimal duration of antibiotic treatment between the two stages, the length of the antibiotic-free period before reimplantation, and the influence of preimplantation culture results on infection eradication, aspects that may be further assessed in future works. Few data exist about the appropriate intravenous-to-oral antibiotic step-down treatment, and in most cases, it is carried out at the discretion of each center. It highlights the importance of a multidisciplinary approach to decision-making.

## 5. Conclusions

Two-stage revision of staphylococcal hip and knee periprosthetic joint infection (PJI), performed with a handmade antibiotic-loaded spacer containing multiple high doses of gentamicin, clindamycin, ceftazidime, and vancomycin, is an effective strategy. It achieves eradication rates of up to 86%. A superimposable outcome is found when systemic rifampicin is not employed, suggesting that high concentrations of local antibiotics serve as an alternative to rifampicin treatment. This is particularly relevant in elderly patients and those at high risk of interaction and adverse effects with this drug. The systemic and mechanical safety of this protocol has also been established. We emphasize the importance of enhancing the elution capacity of handmade spacers with various antibiotics to achieve high concentrations of local antibiotics in the immediate postoperative period. This allows them to act simultaneously through different metabolic pathways. Finally, we would like to highlight the significance of thorough surgical, mechanical, and chemical debridement aimed at eliminating the biofilm and the majority of the latent bacterial inoculum.

## Figures and Tables

**Figure 1 antibiotics-13-00538-f001:**
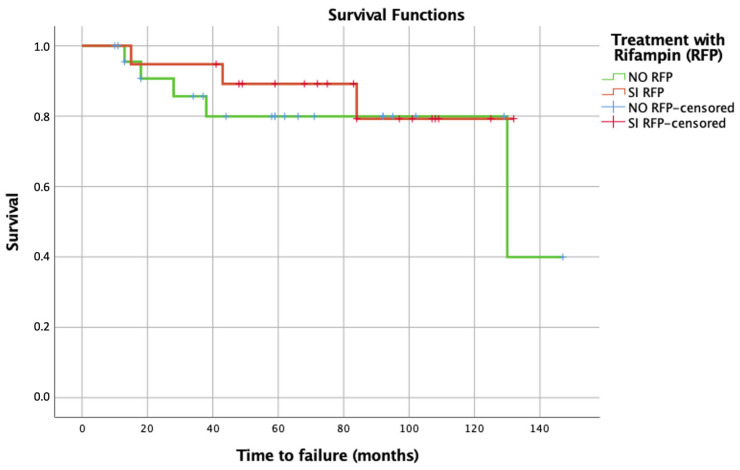
Kaplan–Meier survival analysis for staphylococcal hip and prosthetic joint infection related to antimicrobial treatment strategy followed (rifampicin [RFP] or non-rifampicin regimen). On the X-axis, time is represented in months.

**Figure 2 antibiotics-13-00538-f002:**
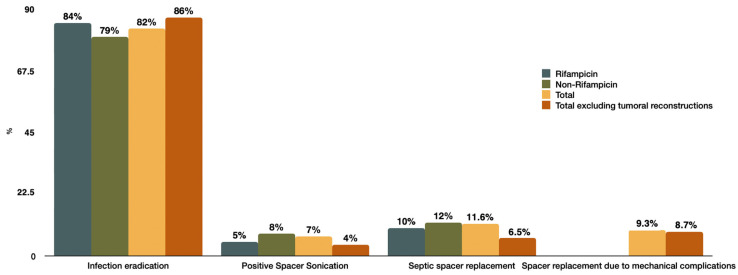
Graphical representation of primary and secondary outcomes as a function of the total sample, excluding PJI secondary to tumoral reconstruction and dependent on rifampicin treatment.

**Figure 3 antibiotics-13-00538-f003:**
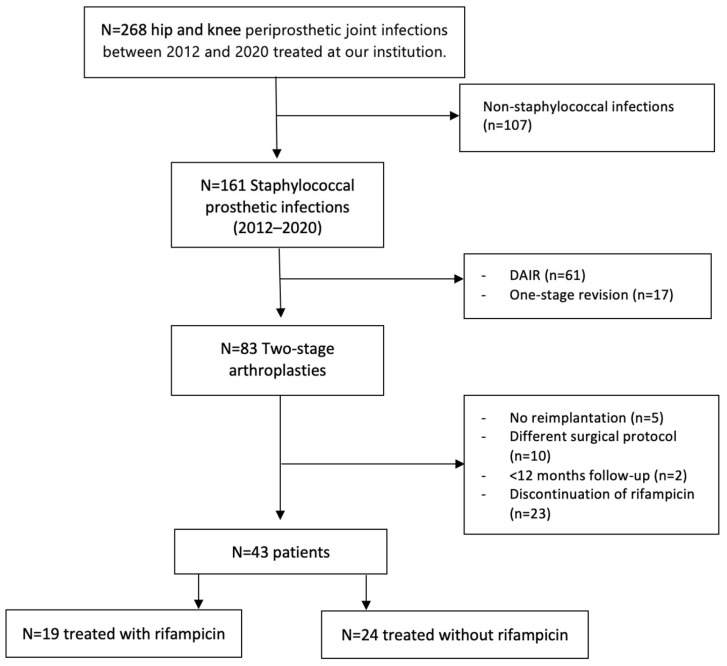
Patient selection algorithm and exclusion criteria. DAIR = Debridement, antibiotics, and implant retainment.

**Figure 4 antibiotics-13-00538-f004:**
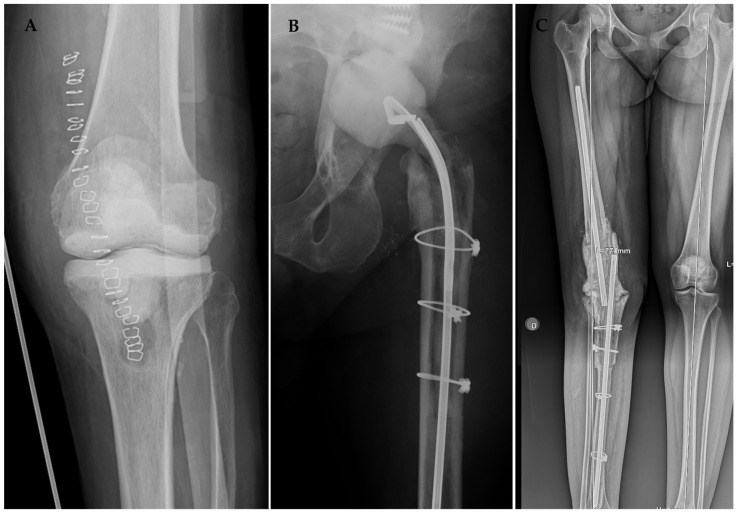
Radiological outcomes of our surgical protocol after the first stage of a knee articulated spacer (**A**), hip articulated spacer (**B**), and static knee spacer (**C**) secondary to an extensive bone defect.

**Table 1 antibiotics-13-00538-t001:** Baseline characteristics of all patients and after stratification for antibiotic treatment strategy (rifampicin [RFP] and non-rifampicin [Non-RFP] cohorts). Univariate and bivariate analysis.

	All	RFP Regimen	Non-RFP Regimen	*p* Value
N patients	43	19 (44%)	24 (55%)	
General characteristics				
Male Sex (%)	18 (42%)	9 (47.5%)	9 (37.5%)	0.516
Age [years, SD ^1^]	70 (±12.5)	66.9 (±14.1)	72.5 (±10.7)	0.276
Time from implantation, [weeks, SD mean]	142 (±235.1)	85.7 (±140.5)	187 (±284.2)	0.167
Origin, another center	15 (35%)	5 (26.3%)	10 (41.7%)	0.294
Comorbidities				
Diabetes	7 (16%)	3 (16%)	4 (17%)	0.911
Chronic kidney disease	5 (11.5%)	0	5 (21%)	0.037
Immunosuppressants	1 (2%)	1 (5%)	0	0.254
Ischemic cardiopathy	6 (14%)	3 (16%)	3 (12.5%)	0.751
Cardiac Insufficiency	3 (7%)	1 (5%)	2 (8%)	0.674
Oral anticoagulants	7 (16%)	2 (10.5%)	5 (21%)	0.631
Prosthesis				
TKA ^2^	24 (55%)	11 (58%)	13 (54%)	0.875
THA ^3^	17 (39.5%)	7 (37%)	10 (42%)	0.714
Hemiarthroplasty	2 (4.5%)	1 (5%)	1 (4%)	0.857
Previous aseptic revision	16 (37.2%)	6 (31%)	10 (41.7%)	0.521
Previous DAIR	7 (16.3%)	2 (10.5%)	5 (20.8%)	0.636
Tumoral reconstruction	6 (14%)	3 (16%)	3 (12.5%)	0.751
Infection characteristics				
Fistula	11 (25.5%)	4 (21%)	7 (29%)	0.563
Tsukayama classification				
I–II	15 (34.8%)	6 (31.5%)	9 (37.5%)	0.913
III	5 (11.5%)	2 (10.5%)	3 (12.5%)	0.851
IV	23 (53.5%)	11 (58%)	12 (50%)	0.677

^1^ SD = standard deviation; ^2^ TKA = total knee arthroplasty; ^3^ THA = Total hip arthroplasty; Tsukayama classification was used to affiliate PJI regarding intraoperative positive cultures (I), early postoperative infection (II) developing < 1 month after index surgery, late chronic infection (III), or acute hematogenous (IV) [19].

**Table 2 antibiotics-13-00538-t002:** Microbiological isolates and resistances of the patients included in the analysis are provided. Additionally, the specific antibiotic treatment of patients with polymicrobial infections is shown.

	All	RFP Regimen	Non-RFP Regimen	*p*-Value	Other Resistances	Antibiotic Treatment
Microorganism	43	19 (44%)	24 (55%)			
*S. aureus*	12 (27.5%)	6 (31.5%)	6 (25%)	0.622	Aminoglycosides (*n* = 1), quinolones (*n* = 1)	
Methicillin-resistant	2 (4.8%)	2 (10.5%)	0	0.107		
Methicillin-sensitive	10 (23.8%)	4 (21%)	6 (25%)	0.769		
*S. epidermidis*	18 (41.8%)	10 (52%)	8 (33.3%)	0.321	Aminoglycosides (*n* = 6), quinolones (*n* = 7), cotrimoxazole (*n* = 3), fosfomycin (1), erythromycin (*n* = 3)	
Methicillin-resistant	11 (25.5%)	6 (31.5%)	5 (20%)	0.622		
Vancomycin-resistant	1 (2.3%)	1 (5.2%)	0			
Rifampicin	1 (2.3%)	0	1			
*S. lugdunensis*	2 (4.8%)	1 (5.2%)	1 (4%)	0.479		Levofloxacin + amoxicillin
*S. capitis*	1	1 (5.2%)	0			Ciprofloxacin + rifampicin
Polymicrobial	10 (23%)	1 (5.2%)	9 (37.5%)	0.013		
MRSA + *S. agalactiae* + *Serratia*	1	1 (5.2%)	0			Cotrimoxazole, clindamycin, rifampicin
MRSA + MSSE	1	0	1 (4%			Cotrimoxazole
MSSA+ *E. cloacae*	1	0	1 (4%)		*E. cloacae:* amoxicillin and ampicillin	Levofloxacin + Fosfomycin
MRSE + *C. albicans*	1	0	1 (4%)		MRSE: Quinolones	Daptomycin + fluconazole
MRSE + SL + *P. mirabilis*	1	0	1 (4%)			Ciprofloxacin
MRSE + *E. feacalis* + *E. faecium*	1	0	1 (4%)		*Enterococci:* gentamicin, tobramycin, amikacyn, ciprofloxacin, cotrimoxazole	Ciprofloxacin + cotrimoxazole
MRSE + *E. feacalis*	1	0	1 (4%)		*Enterococci:* aminoglycosides, cotrimoxazole and quinolones	Tedizolid
MRSE + MSSA	1	0	1 (4%)		*MRSE*: aminoglycosides, erythromycin, clindamycin, cotrimoxazole, quinolones	Linezolid + Fosfomycin
MSSE + *H. parainfluenzae*	1	0	1 (4%)		*MSSE*: clindamycin, erythromycin	Amoxicillin + Levofloxacin
SL + *C. parapsilosis*	1	0	1(4%)			Caspofungin + Cotrimoxazole + Levofloxacin

MRSA = Methicillin-resistant *Staphylococcus aureus;* MSSA = Methicillin-sensitive *Staphylococcus aureus*; MRSE = Methicillin-resistant *Staphylococcus epidermidis*; MSSE = Methicillin-Sensitive *Staphylococcus aureus*; SL = *Staphylococcus lugdunensis.*

**Table 3 antibiotics-13-00538-t003:** Follow-up and treatment outcomes of all patients, and after stratification for antibiotic treatment strategy (rifampicin [RFP] and non-rifampicin [Non-RFP]). Univariate and bivariate analysis.

	All	RFP Regimen	Non-RFP Regimen	*p* Value
N patients	43	19	24	
Duration of antibiotic treatment [weeks, SD ^1^]	11.5 (±5.9)	11.8 (±5.8)	11.4 (±6.1)	0.738
Surgical management				
Articulated spacer	31 (72%)	15 (79%)	16 (67%)	0.373
Spacer exchange (%)	9 (21%)	5 (26%)	4 (16%)	0.440
Mechanical complication	4 (9.3%)	3 (16%)	1 (4%)	0.613
Infection persistence	5 (11.6%)	2 (10.5%)	3 (12.5%)	0.841
Time to reimplantation [weeks, SD]	14.6 (±5)	13 (±5.2)	15.7 (±4.6)	0.079
Time of hospitalization [days SD]	58 (±61.2)	37 (±28.2)	76 (±75.5)	0.219
Medical complication	18 (42%)	6 (32%)	12 (50%)	0.224
Surgical complication	11 (25%)	6 (31.5%)	8 (33%)	1.000
Second stage				
Spacer sonication (+)	3 (7%)	1 (5%)	2 (8.3%)	0.695
Deep tissue cultures (+)	8 (18.5%)	4 (16.7%)	4 (21%)	0.714
Treatment failure ^2^	8 (18%)	3 (16%)	5 (21%)	0.673
Death	0	0	0	
Reintervention	7 (16%)	3 (16%)	4 (16%)	
Suppressive therapy	1 (2%)	0	1 (4%)	
Responsible microorganism				
Reinfection	5 (11%)	2 (10%)	3 (12.5%)	0.841
Superinfection	3 (7%)	1 (5.2%)	2 (8.3%)	0.721
Treatment failure excluding tumoral reconstructions	5 (13.5%)	2 (12.5%)	3 (14%)	0.875
Follow-up time, [months, SD]	68 (±37.7)	79 (±31.2)	59.4 (±40.7)	0.293

^1^ SD = standard deviation; ^2^ Treatment failure (Not cured) is defined by: (a) reintervention because of septic reasons (persistence of infection, reinfection, or superinfection); (b) PJI-related death; (c) suppressive antibiotic therapy.

**Table 4 antibiotics-13-00538-t004:** Univariate and Multivariate analyses of all variables investigated as predictors of treatment failure.

Covariables	Univariate Analysis			Multivariate Analysis		
	OR	95% CI	*p*-Value	OR	95% CI	*p*-Value
Patient characteristics						
Age			0.188 ^1^	1.02	0.91–1.13	0.736
Time from implantation			0.095 ^1^	1.01	0.99–1.04	0.071
TKA	1.4	0.3–6.8	0.673	Ref.		
THA	0.4	0.07–2.5	0.351	1.98	0.20–18.7	0.551
Previous revision	0.5	0.01–2.8	0.428			
Previous DAIR	2.6	0.3–19.7	0.459			
Tumoral reconstruction	6.4	0.9–41	0.033	1.23	0.33–45.5	0.911
Diabetes	1.6	0.2–10.3	0.731			
Oral anticoagulants	0.7	0.01–6.7	0.587			
Infection characteristics						
All pathogens	Ref.			Ref.		
MSSA	0.3	0.03–3.1	0.084			0.999
MRSA	4.1	0.2–73.2	0.659			
CNS	2.5	0.5–12.1	0.248	1.64	0.18–14.98	0.661
Monomicrobial infection	Ref.			Ref.		
Polymicrobial infection	4.8	0.9–24.9	0.047	14.62	1.05–203.22	0.043
Treatment						
Rifampicin regimen	Ref.			Ref.		
No rifampicin use	1.3	0.3–7.8	0.673	1.99	0.10–36.71	0.693
Time to reimplantation			0.208 ^1^	1.13	0.89–1.43	0.299
Static spacer	Ref.			Ref.		
Articulated spacer	0.3	0.06–1.4	0.123	0.11	0.01–1.22	0.071
Septic spacer exchange	9.9	1.3–74.7	0.011	14.25	1.42–142.1	0.024

^1^ *p*-value for continuous variables with non-normal distribution was calculated with the Mann–Whitney U test.

**Table 5 antibiotics-13-00538-t005:** Surgical complications depicted depending on the type of spacer.

	All Spacers	Hip	Knee		*p* Value ^1^
			Static	Articulated	
N patients	43	19 (44.2%)	7 (16.3%)	17 (39.5%)	
Mechanical major complication	6 (14%)	5 (26%)	0	1 (5.8%)	0.072
Spacer dislocation	4 (9.3%)	4 (21%)	0	0	
Fracture of the spacer	1 (2.3%)	1 (5.2%)	0	0	
Intraoperative fracture	1 (2.3%)	0	0	1 (5.8%)	
Wound dehiscence	5 (11.6%)	1 (5.2%)	2 (28.5%)	2 (11.7%)	0.618
Other non-septic reintervention ^2^	5 (11.6%)	1 (5.2%)	0	4 (23.5%)	

^1^ *p*-value calculated with Fisher’s test for H_o_ of no differences between knee spacer and hip spacer. ^2^ Other non-septic reinterventions included vascular complications, arthrolysis, and peri-implant fracture through the follow-up period.

## Data Availability

Data are unavailable due to privacy restrictions.

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
