# Peer review of "Do High Doses of Multiple Antibiotics Loaded into Bone Cement Spacers Improve the Success Rate in Staphylococcal Periprosthetic Joint Infection When Rifampicin Cannot Be Employed?"

_antibiotics, 2024, doi:10.3390/antibiotics13060538_

Round 1

Reviewer 1 Report

Comments and Suggestions for Authors

The authors present a retrospective single-center observational study trying to evaluate the use of multiple-high-dose antibiotic-loaded cement spacer in treatment of PJI with 2-stage revision.

This is a very interesting study that describes a novel antibiotic cement spacer in treatment of PJI and especially in difficult to treat PJIs such as those caused by rifampicin-resistant bacteria.

My main issue with the manuscript is that the authors are focusing mostly to compare the results within the 2 groups of rifampicin-sensitive and rifampicin-resistant bacteria caused the PJI.

In my opinion the difference between the 2 groups is an interesting result but it cannot be the main conclusion and outcome of the study. That is due to at least 2 reasons: 

- the groups are very heterogenous with very few patients in order to get any conclusions. On the other hand, the study give us an indication that the use of the novel MHDALBC spacer can be useful in difficult to treat cases like those caused by rifampicin-resistant bacteria.

- in 2 stage revision all the foreign material (infected prostheses) are extracted and there is no indication on using biofilm-active antibiotics such as rifampicin. In the most popular and established PJI-treatment protocols rifampicin is not used for the treatment between the first and the second stage of the 2-stage PJI revision. The authors mention also that in the introduction part of the manuscript (lines 52-59).

I believe that the manuscript provides important information on the role of the MHDALBC spacer in treatment of PJI and the results have to presented by looking on the whole group of patients (a treatment success of 82%!). A further analysis looking especially on the difficult to treat PJI (such as those caused by rifampicin-resistant bacteria) can be done and discussed also.  

Also, some other comments:

-       Lines 174-179: was rifampicin used on these patients where the cultures were positive in second stage?

-       Lines 183-190: Please define if the spacer exchange due to persistent infection was counted as a failure. Was it counted as a failure even though after the cement spacer exchange the treatment was successful? 

-       Lines 183-190: Should a positive spacer culture be counted as a failure? Explain. There is big risk of contamination in spacer sonication (false positive results – not clinically relevant).

-       In all tables, please define how the time is reported (months/weeks/days?) and try to be consist with the manuscript text.

-       Table 1: Can you report the number of cases that were infected after arthroplasty due to trauma? We report 2 cases of hemiarthroplasty which means that the arthroplasty was performed after cervical fracture. How many were those cases and how many were after elective surgery. The trauma cases have per definition higher risk of infection due to trauma itself but also due to elderly and more sick patients.

-       Table 1 and lines 93-95: Is it necessary to report the Tsukayama classification? It does not add many to the analysis. You can just report as early, chronic and hematogenous. If you insist, so please report together the typ I and II (only one case in type I). 

-       Lines 162-165: Does a mechanical complication count as a failure? If not how do you manage these cases. The spacer exchange due to mechanical complications hides high risk of new infection.

-   Lines 370-374: it is well known that tumoral reconstructive surgeries have much higher infection rates due to various reasons. These are not common cases I clinical praxis and may consider exclude from the patient cohort. That would lead to a 86% cure of the cases remaining! 

Reviewer 2 Report

Comments and Suggestions for Authors

Antibiotics-2976825 Do high doses of multiple antibiotics loaded to bone cement spacers improve success rate in staphylococcal periprosthetic joint infection when rifampicin cannot be employed?

In the manuscript the authors describe the investigation of the clinical and microbiological outcomes of patients subjected to a hip or knee arthroplasty’s revision. Even if the topic reported in the research is current however there are several previously published paper reporting the same types of analysis thus the novelty of the manuscript not fully emerged.

 Here, only few manuscripts related to the topic published:

 1)      López-Torres II, Vaquero-Martín J, Torres-Suárez AI, Navarro-García F, Fraguas-Sánchez AI, León-Román VE, Sanz-Ruíz P. The tale of microencapsulated rifampicin: is it useful for the treatment of periprosthetic joint infection? 2022

2)      Laperche J, Barrett CC, Boduch A, Glasser J, Clippert D, Garcia DR, Antoci V. Mechanically stable rifampin antibiotic cement inhibits Pseudomonas aeruginosa biofilm surface growth. 2024

3)      Gálvez-López R, Peña-Monje A, Antelo-Lorenzo R, Guardia-Olmedo J, Moliz J, Hernández-Quero J, Parra-Ruiz J. Elution kinetics, antimicrobial activity, and mechanical properties of 11 different antibiotic loaded acrylic bone cement. 2014

4)      Sanz-Ruiz P, Carbó-Laso E, Del Real-Romero JC, Arán-Ais F, Ballesteros-Iglesias Y, Paz-Jiménez E, Sánchez-Navarro M, Pérez-Limiñana MÁ, Vaquero-Martín J. Microencapsulation of rifampicin: A technique to preserve the mechanical properties of bone cement. 2018

5)      Trombetta RP, Ninomiya MJ, El-Atawneh IM, Knapp EK, de Mesy Bentley KL, Dunman PM, Schwarz EM, Kates SL, Awad HA. Calcium Phosphate Spacers for the Local Delivery of Sitafloxacin and Rifampin to Treat Orthopedic Infections: Efficacy and Proof of Concept in a Mouse Model of Single-Stage Revision of Device-Associated Osteomyelitis. 2019

 In addition, it is very poor in several aspects, described as follows:

Major Concerns

1)      The introduction section is confused and no logic displaying of the concepts is followed. In lines 50-59 the authors reported that Staphylococcus aureus and S. epidermidis are the most common pathogens involved in PJIs, however no data of their recovery is reported and no mention related to their resistant profiles. Additionally, lines 63-65 and 75-79 are also not clear in their meaning. The aim of the work is to be rewrite since no literature explaining the novelty and the need of such study is reported (lines 80-84). 

 2)      In the material and methods, the authors stated that the patients were selected between 2012-2020: since we are in 2024 I strongly suggest to enlarge the period at least until 2022-2023. Which is the “single center” (line 87) involved in the study, it should be mentioned. Additionally, no information on how the spacers were made is provided, neither their antibiotic’s elution capability ((lines 447-450). For the microbiological point of view, it is not sufficient what the authors described in lines 120-123: where the samples were inoculated? How the microbiological analysis was ruled out? How was made the strain identification? And their resistant pattern identified? Just as few examples.

 3)      In the results section, lines 228-234, the authors briefly reported the microbiological analysis. The resistant pattern of the isolates should be described as it is necessary to specify other CoNS which ones? Polymicrobial, sustained by which microorganisms? To accomplish this request would be suitable split table 1 in two tables, one with patients’ characteristics and the other with all the microbiological results. Figure 2 should be re-designed since is not easily understandable and also some mistakes are also present. In lines 267-270 the authors stated that recurrence and superinfections occurred but no data regarding this issue are reported in the section. These data should be provided. In lines 295-299 other microbiological results are provided but it is not clear why they are reported here and not in the previous

 4)      In the discussion section the authors stated that “the best combination of antimicrobials and specific dosage is yet unknown” (lines 333-334) but no references are reported to corroborate this statement.  In lines 344-346 the authors did not suggest an explanation for the reasons of the variability in the success rates, and also in lines 365-366 regarding the enrolled population. In lines 348-354 different antibiotics and their profiles are reported but it is not clear the connection with the paper’s topic: rephrased it. In lines 354-356 a comparison between in vitro and in vivo studies is reported but it is not very fair to compare these kinds of results that are so different in the methodologies and background. In lines 382-384 and 389-392 the authors reported that biofilm is a key factor for PJIs, however they did not perform the biofilm test on the staphylococcal isolates and I strongly recommend to complete also these experiments.

 5)      The conclusion and abstract have to be rewritten considering the above-mentioned major points.

 6)      The references are really dated and only 14/72 are in/after 2020. I suggest to eliminate the oldest to give place to a more recently literature.

Minor Concerns.

The authors used abbreviation improperly and Staphylococcus nomenclature too.

Comments on the Quality of English Language

The text of the manuscript needs significant English Editing

There are many typos in the text that reflect superficial manuscript preparation

Reviewer 3 Report

Comments and Suggestions for Authors

1.     This article explored the efficacy and safety of a novel two-stage revision protocol for staphylococcal PJI in patients un-20 suitable for rifampicin therapy, based on the use of specific multiple high-dose antibiotics-loaded 21 bone cement spacer.

2.     This topic is original in the field, it provides a viable and safe alternative for patients ineligible for rifampicin treatment.

3.     The tables and results can be improved to emphasize your main conclusions.

4.     Is there any limitation of this study? How about the patient sizes to meet some kinds of standard?

5.     The conclusions are consistent with the current evidences. However, It should be raveled out, in other words, more readable.

6.     The references are roughly appropriate.

Round 2

Reviewer 1 Report

Comments and Suggestions for Authors

Thank you for addressing my points of criticism.

No additional comments.

Kind regards,

Stergios Lazarinis 

Author Response

Thank you for your comments to improve our manuscript

Reviewer 2 Report

Comments and Suggestions for Authors

Antibiotics-2976825 Do high doses of multiple antibiotics loaded to bone cement spacers improve success rate in staphylococcal periprosthetic joint infection when rifampicin cannot be employed?

The authors made efforts to improve the overall quality of the paper since many modifications were made in the manuscript text, addressing further comments in the letter.  In any case, the reviewer considers it necessary for the authors to implement both the cited references and the English of the text. In fact, as previously stated in the letter: “The references are really dated and only 14/72 are in/after 2020. I suggest to eliminate the oldest to give place to a more recently literature.

Comments on the Quality of English Language

In the text there are still typos and sentences to rewrite

Author Response

  • We removed the oldest references, going from 72 to 49 references
  • We have sent the manuscript to special company to review the English grammar as reviewer recommended.